# GLP-1 Receptor Agonists in Non-Alcoholic Fatty Liver Disease: Current Evidence and Future Perspectives

**DOI:** 10.3390/ijms24021703

**Published:** 2023-01-15

**Authors:** Riccardo Nevola, Raffaella Epifani, Simona Imbriani, Giovanni Tortorella, Concetta Aprea, Raffaele Galiero, Luca Rinaldi, Raffaele Marfella, Ferdinando Carlo Sasso

**Affiliations:** Department of Advanced Medical and Surgical Sciences, Università Della Campania “Luigi Vanvitelli”, I-80131 Naples, Italy

**Keywords:** NAFLD, GLP-1 receptor agonist, type 2 diabetes, steatosis

## Abstract

To date, non-alcoholic fatty liver disease (NAFLD) is the most frequent liver disease, affecting up to 70% of patients with diabetes. Currently, there are no specific drugs available for its treatment. Beyond their anti-hyperglycemic effect and the surprising role of cardio- and nephroprotection, GLP-1 receptor agonists (GLP-1 RAs) have shown a significant impact on body weight and clinical, biochemical and histological markers of fatty liver and fibrosis in patients with NAFLD. Therefore, GLP-1 RAs could be a weapon for the treatment of both diabetes mellitus and NAFLD. The aim of this review is to summarize the evidence currently available on the role of GLP-1 RAs in the treatment of NAFLD and to hypothesize potential future scenarios.

## 1. Introduction

Due to the spread of metabolic syndrome, non-alcoholic fatty liver disease (NAFLD) has become the most frequent liver disease worldwide, reaching a global estimated prevalence of 32.4% [1]. Characterized by an excessive accumulation of intra-hepatocyte lipids, it can be complicated by the development of inflammation and fibrosis, evolving towards forms of non-alcoholic steatohepatitis (NASH) and liver cirrhosis [2]. In light of the close etiopathogenetic correlation with metabolic syndrome, NAFLD has been recently redefined as metabolic dysfunction-associated fatty liver disease (MAFLD) [3]. NAFLD and the cornerstones of metabolic syndrome (type 2 diabetes mellitus—T2DM, visceral obesity, arterial hypertension, dyslipidemia) share a common pathogenesis, closely linked to resistance to insulin action [4]. The insulin resistance in T2DM and NAFLD determines an increase in hepatic production and reduced muscle glucose uptake, as well as lowering lipolysis in adipose tissue and increasing hepatic lipogenesis, consequently raising glucose and circulating free fatty acids (FFAs) levels [5]. The excess of circulating FFAs results in their increased flow in the liver and their storage, mainly in the form of triglycerides [4]. The high levels of circulating FFAs are the cause of a further worsening of the insulin resistance degree through the inhibition of the post-receptor insulin signal, fueling a vicious cycle [6].

To date, the only treatment proved to be effective in NAFLD is a weight loss of ≥7–10%, able to significantly improve the histological grade of fibrosis and steatosis, in addition to hepatic cytolysis [7,8]. The use of pioglitazone [9] and high doses of vitamin E [10] have demonstrated some effectiveness at the expense of potential serious adverse effects. However, in clinical practice, no pharmacological treatments that could support the necessary lifestyle changes (reduced caloric intake and increased physical activity) are available. Moreover, the use of obeticholic acid [11] has not been recommended by regulatory agencies yet.

The glucagon-like peptide-1 (GLP-1) is a peptide hormone produced by gut enteroendocrine cells that acts on insulin and glucagon secretion, as well as food intake and gut motility, regulating postprandial glucose homeostasis [12]. The receptor agonists of GLP-1 (GLP-1 RAs) are able to increase the hyperglycemia-induced insulin secretion and decrease glucagon secretion, delay gastric emptying, reduce appetite and caloric intake, and they are strongly recommended in T2DM treatment [13]. Similarly to sodium–glucose linked transporter-2 (SGLT2) inhibitors [14], GLP-1 RAs have demonstrated a surprising cardio and nephroprotective action since they significantly reduce major adverse cardiovascular events’ (MACEs) rate and the risk of kidney disease progression in patients with T2DM [15]. Due to their significant contribution to weight loss, they have been recently approved for the treatment of obesity in patients with or without T2DM [16,17]. Growing evidence is building on the efficacy of GLP-1 RAs in NAFLD. Through a multifactorial and not completely understood mechanism of action, they seem to improve liver damage, favoring a reduction in the degree of steatosis, inflammation and hopefully liver fibrosis. The magnitude and the long-term clinical impact of these effects is still being evaluated, but the data available suggest GLP-1 RAs (and likely SGLT2 inhibitors) play a main role in the treatment of NAFLD.

This review aims to collect all the current knowledge on the role of GLP-1 RAs in NAFLD and to describe their potential impact on the course of liver disease and its complications, evaluating future clinical use scenarios. In this review, we will keep the nomenclature of NAFLD (instead of MAFLD) since in most of the cited studies, patient enrollment was based on previous diagnostic criteria.

## 2. GLP-1 Receptor Agonists: Mechanisms of Action and Current Indications

GLP-1 is an incretinic hormone of 30 amino acids whose action is mediated by a specific receptor (GLP-1 receptor, GLP-1R) located in the bowel (distal ileum and colon), in the pancreatic alpha and beta cells and in the central nervous system (CNS) [18]. Even though at lower levels, it is also expressed in the heart, lungs, kidney, in blood vessels and peripheral nervous system [18,19]. The main effect of GLP-1 is the enhancement of insulin secretion from beta-cells in response to hyperglycemia and the inhibition of the release of glucagon from alpha cells [20]. Furthermore, GLP-1 slows gastric emptying and gastrointestinal motility to reduce glucose absorption, allowing the control of post-prandial glucose and triglycerides excursions. In addition to the mechanisms on intestinal transit, GLP-1 is also able to induce satiety through a direct stimulation of CNS [19].

Through these mechanisms of action, GLP-1 RAs (exenatide, lixisenatide, liraglutide, dulaglutide, albiglutide, semaglutide) strengthen the physiological effect of endogenous GLP-1. Overall, these drugs are able to lower glucose (mainly postprandial) and glycosylated hemoglobin blood levels, simultaneously promoting weight loss. Compared to the original peptide, which is rapidly degraded by enzyme dipeptidyl peptidase-4 (DPP-4) in 1.5 min [21], GLP-1 RAs show a long-term action in resisting the degradation through DPP-4. The addition of other molecules (e.g.: fatty acid, immunogloblulin, albumin) allowed the daily (liraglutide, semaglutide oral) or weekly (dulaglutide, albiglutide e semaglutide subcutaneous) administration [22]. The chronic treatment with GLP-1 RAs is able to reduce glycosylated hemoglobin from 0.8% to 1.9%, in relation to molecule and dosage used [23]. On average, long-acting agents lead to a greater reduction in glycosylated hemoglobin compared to short-acting agents given the better post prandial and nocturnal glycemic control [24]. Taking into account the considerable reduction in incretin effect (i.e., the secretion of insulin in response to oral glucose ingestion mediated by GLP-1 and glucose-dependent insulinotropic peptide—GIP) in patients with T2DM and the significant anti-hyperglycemic action, GLP-1 RAs are strongly recommended in the management of patients with T2DM [13]. 

In addition to the glycemic control in T2DM, GLP-1 RAs have been demonstrated to be extraordinarily effective in the prevention and treatment of its complications [24]. In particular, they have been shown to effectively reduce the rate of MACEs (such as myocardial infarction and ischemic stroke) and related mortality [25,26,27]. Overall, GLP-1 RAs treatment is associated with a 9–16% reduction in cardiovascular events and all-cause mortality [24]. For this reason, GLP-1 RAs have become first choice drugs in the subject of diabetes with evidence of atherosclerotic disease and/or previous cardiovascular event [13]. In addition, GLP-1 RAs seem to prevent the progression of T2DM-associated renal damage [24]. Treatment with these drugs is in fact associated with a reduced decline in the estimated glomerular filtration rate (eGFR), as well as with a reduction in urinary albumin excretion and in the incidence rate of de novo albuminuria [28,29]. However, in this setting, more solid evidence and a greater effect have been shown for the SGLT-2 than for GLP-1 RAs [15].

Since GLP-1 RAs are able to reduce appetite and induce satiety and early post-prandial fullness, their use is associated with a significant weight loss secondary to a reduced caloric intake [24]. The effect of treatment with GLP-1 on body weight accounts for a decrease from 1.5 to 6 kg, in relation to the molecule and dosage, with greater effect for semaglutide [23]. This anorectic action allowed GLP-1 RAs to be approved for the treatment of obesity, also in the absence of T2DM [16,17].

Overall, the favorable metabolic profile induced by GLP-1 RAs (weight loss and reduced caloric intake, improvement in glycemic compensation) allows one to hypothesize their potential effectiveness in the treatment of NAFLD, in addition to T2DM and obesity. Growing evidence has been produced in this setting. GLP-1 RAs was shown to positively impact steatosis and liver inflammation and potentially fibrosis degree. To date, it is not known if the benefits of treatment with GLP-1 RAs are direct or mediated by drug-induced weight loss.

## 3. Effects of GLP-1 Receptor Agonists on NAFLD

### 3.1. Effects of GLP-1 RAs on Body Weight

To date, the only validated treatment for the management of NAFLD is weight loss, generally achieved through lifestyle changes (diet, physical activity) or bariatric surgery. Weight loss of ≥7–10% has been shown to reduce disease progression and improve steatosis, inflammation, and liver fibrosis [7,8]. However, a significant and long-lasting reduction in body weight is difficult to achieve and various dietary regimens often fail both in the long term and in highly motivated patients. Moreover, counter-regulatory mechanisms over time hinder weight loss, resulting in the recovery of body weight [30]. In addition to lifestyle changes, GLP-1 RAs have been shown to lower body weight in a range of 2–7 kg through early satiety and reduction in appetite, thus lowering caloric intake [31,32,33,34,35,36,37,38,39,40,41,42,43,44,45,46,47,48,49,50]. Furthermore, treatment with these drugs reduces cravings for food and increases the pleasure offered by eating a meal [51]. Except for SGLT2 inhibitors, all the other hypoglycaemics drugs (including insulin) either have neutral effect or, more often, cause weight gain [24].

Mechanisms involved in this process are both central and peripheral. In the CNS, serotonin represents a crucial factor in appetite regulation. Through the stimulation of its receptor in the central nervous system, GLP-1 reduces the expression of serotonin 5-HT2A receptors in the hypothalamus, resulting in decreased appetite [52]. This role is independent from serotonin secretion, which is not affected [53]. In particular, in murin models, GLP-1 RAs have demonstrated to prevent meal-initiation-inhibiting activity on arcuate nucleus and inducing meal-ending actions on lateral parabrachial nucleus [24]. It has been hypothesized that liraglutide and semaglutide act through different neurological pathways, providing the rationale for their different efficacy in inducing weight loss [54]. The reduction in appetite is also assisted by peripheral mechanisms. In fact, GLP-1 RAs are able to delay gastric emptying and to slow intestinal motility in the gastro-intestinal tract, which enhances the feeling of early satiety at a central level [45]. However, the long-term receptor stimulation by GLP-1 over time reduces the effects on gastric emptying due to the mechanisms of tachyphylaxis [55].

The main studies on the efficacy of GLP-1 RAs in inducing weight loss are summarized in Table 1. The extent of the reduction in body weight depends on which GLP-1 RAs and which dosage are used. Moreover, the dose used to treat T2DM and obesity may differ. In fact, if the glycemic control plateau is reached with relatively low dosages, weight loss effects may require higher GLP-1 RAs dosages [24]. In contrast, the variability in interindividual response on body weight loss in patients receiving GLP-1 RAs treatment is significantly greater than the variability in glycemic control response [16]. To date, no specific predictors of long-term response to weight loss have been identified.

Liraglutide has demonstrated to determine significant weight loss, reduce pre-diabetes rate and improve obesity-related risk factors independently from the presence of T2DM [38,40,41,50,56,57,58,59]. After a year of treatment at the daily dosage of 3 mg, it was able to reduce body weight by 8.4 kg [33]. Comparing the results from the various trials [33,38,42,43,44,47,48], the average weight loss is about 3.4–6.1% of the total body weight. In particular, among treated patients, about 2/3 show a weight loss of at least 5% and 1/3 a weight loss of more than 10% [33]. Its anorectic effect starts very early; therefore, in mice, it has significantly reduced appetite within 1 h of administration [53]. To our knowledge, only one study failed to prove significant differences in weight loss between treatment with liraglutide and placebo [60].

Semaglutide is a 2nd generation GLP-1 agonist available both in oral (daily administration) and subcutaneous (weekly administration) formulation [23,24]. The treatment with semaglutide resulted in a lower energy intake and greater weight loss than placebo both in adults [31,34,35,36,37,39,61] and adolescents [49]. In particular, at a dosage of 2.4 mg per week, it has been shown to induce an average weight loss between 9.6% and 17.4%, as well as a reduction in waist circumference and arterial hypertension and an improvement of lipid profile [34,35,36,37,39]. The dose used for the treatment of obesity (2.4 mg/week) is significantly more effective in determining weight loss than that used for the treatment of T2DM (1.0 mg/week) [35]. Furthermore, the ability to induce weight loss is more pronounced for semaglutide than other GLP-1 RAs [32]. For doses equal to or greater than 0.2 mg/day (1.4 mg/week), semaglutide is also superior to liraglutide (at any dose) in causing weight loss [31]. Notably, in the recent STEP 8 randomized clinical trial [46], semaglutide (2.4 mg/week) achieved a mean weight loss of 15.8% after 68 weeks of treatment compared with 6.4% obtained with liraglutide. A weight loss greater than 10% (therapeutic target in NAFLD) was achieved in 70.9% of semaglutide-treated patients and 25.6% of liraglutide-treated patients, with comparable rates of gastrointestinal adverse events [46]. Semaglutide-related weight loss continued until weeks 28 to 44 of treatment and was sustained thereafter [61].


For these reasons, beyond T2DM, liraglutide and semaglutide have also been approved for the treatment of obesity in diabetic and non-diabetic patients [16,17]. The maximum approved dose of liraglutide for the treatment of obesity is 3 mg/day, whereas the maximum dose for the treatment of T2DM is 1.8 mg/day. The maximum dose of semaglutide used for the treatment of obesity is 2.8 mg/week (0.4 mg/day), whereas the maximum dose for the treatment of T2DM is 1 mg/week.

### 3.2. Effects of GLP-1 RAs on Hepatic Cytolysis

Although not highly sensitive [62], hepatic cytolysis markers (aspartate aminotransferase—AST, and, above all, alanine aminotransferase—ALT) represent the epiphenomenon of the potential hepatic inflammation in NAFLD patients. Indeed, patients with NAFLD and elevation of liver cytolysis enzymes have an increased risk of NASH and progression to cirrhosis [63]. Meanwhile, treatments that reduce the risk of NAFLD progression are all associated with a significant reduction in hepatic cytolysis [7,63,64,65]. In overweight or obese patients with NAFLD, reduction in body weight ≥7–10% improves liver enzymes together with the histological grade of disease [66,67]. Despite normalization of liver injury indices is an independent predictor of fibrosis improvement in patients with NASH [65], for several drugs this improvement is not associated with a significant histological and prognostic benefit [7,68]. Growing evidence supports the positive impact of GLP-1 RAs in reducing hepatic cytolysis enzymes [56,61,69,70,71,72,73,74,75,76,77,78,79,80,81] (Table 2). Buse [69] and Klonoff [70] et al. first showed that long-term treatment with exenatide in patients with T2DM was associated with a significant improvement in biomarkers of liver injury, as well as a reduction in body weight, blood pressure and glycosylated hemoglobin levels. Notably, approximately 40% of patients with baseline liver enzyme elevations achieved normal ALT after treatment. Subsequently, Fan et al. [72] compared treatment with exenatide and metformin in diabetic patients with NAFLD. The authors showed that a 12-week treatment with exenatide was associated with a significant reduction in liver enzymes compared to treatment with metformin. Similarly, Sathyanarayana et al. [71] have also demonstrated that treatment with exenatide, as well as pioglitazone, improves liver injury rates in patients with T2DM. However, the reduction in ALT levels was significantly greater following combined treatment with pioglitazone and exenatide. A similar impact on biomarkers of liver damage has also been demonstrated for lixisenatide [75], dulaglutide [78,79], liraglutide [73,76,77] and semaglutide [61,80]. In patients with T2DM and NAFLD, dulaglutide has been shown to significantly reduce levels of liver injury biomarkers compared to placebo at 6 months of treatment [79]. As concerns liraglutide, Armstrong et al. [73] performed a meta-analysis of 4442 patients with T2DM. At a dosage of 1.8 mg/day, liraglutide has been proven to significantly lower ALT levels in patients with elevated baseline values, with a dose-dependent effect. In fact, no significant differences in the impact on biomarkers of liver injury were observed at the dosage of 0.6 and 1.2 mg/day compared to placebo. Changes in liver fat content would be directly related to the reduction in body weight, serum ALT and triglyceride levels [77]. Finally, in a post hoc analysis of two large trials on the effect of semaglutide on body weight and cardiovascular outcomes, this drug was shown to significantly reduce ALT levels in patients with T2DM and/or obesity [80]. The degree of improvement in the indices of liver damage was directly related to the dosage of semaglutide and greater for high doses. Maximal reduction in ALT levels was achieved approximately 28–30 weeks after initiation of treatment, with subsequent stabilization. These effects have been shown to be closely related to drug-induced weight loss. Recently, Newsome et al. [61] confirmed the efficacy of semaglutide in determining an improvement in hepatic cytolysis indices with a dose-dependent effect.

Another target in the treatment of NAFLD is represented by the reduction in triglycerides’ accumulation in the liver. In this regard, GLP-1 RAs have been proven to significantly improve hepatic steatosis. Preclinical studies showed that exendin-4, a GLP-1R agonist derived from the saliva of the Gila monster, was able to significantly reduce the hepatic accumulation of triglycerides both in mouse models [82] and in human hepatocytes [83] by activating the signaling cascade downstream of the insulin receptor (see below). Furthermore, the addition of exenatide to pioglitazone (already known for its benefits on NAFLD) was associated with a greater reduction in hepatic fat content compared to pioglitazone monotherapy [71].

Although inconsistent data is available [84,85], most studies subsequently confirmed the in vivo benefit of GLP-1 RAs on fatty liver disease [50,56,57,58,61,71,74,76,77,81,84,85,86,87,88,89,90] (Table 3). Compared with other hypoglycemic agents, exenatide significantly reduces the hepatic triglyceride content and epicardial adipose tissue related to weight loss [88]. Braslov et al. [74] underlined that the improvement of hepatic fat accumulation (assessed using the fatty liver index—FLI) induced by exenatide is not detectable for the other oral hypoglycemic agents. In addition to GLP-1 RAs, a similar impact on steatosis (assessed by FLI) was also showed by SGLT2i [81].

Vanderheiden [89], Petit [57] and Yan [50] et al. demonstrated through magnetic resonance imaging a significant reduction in hepatic and subcutaneous fat in patients treated with liraglutide. Treatment with liraglutide would reduce liver fat content by 44% [58]. In addition to the effect on liver fat, liraglutide treatment is also associated with a greater reduction in total fat mass than treatment with sulfonylurea [59,77]. In this setting, the study by Armstrong et al. is very interesting [56]. Through a multicenter, double-blind, placebo-controlled phase 2 trial, the authors evaluated the impact of liraglutide on liver histology in patients with NASH. Treatment with liraglutide (1.8 mg/day) resulted in a greater improvement in steatosis and hepatocyte ballooning compared to placebo. Particularly, liraglutide has been shown to achieve histological resolution of steatohepatitis in 39% of cases after 48 weeks of treatment, compared to 9% of cases in the placebo group. Patients treated with liraglutide had a relative risk of 4.5 (95% CI: 1.1–18.9; *p* = 0.017) of achieving resolution of NASH compared with placebo, regardless of the presence of T2DM. However, NAFLD activity score (NAS) was not significantly different between the two groups, although there may be additional benefits associated with liraglutide dosages >1.8 mg/day, not evaluated in this trial. No significant differences in weight loss and improvement in glycosylated hemoglobin levels were found between liraglutide responders and non-responders. A similar effect of liraglutide in patients with NASH was also highlighted by Eguchi et al. [76]. Recently, the effects of semaglutide treatment on patients with biopsy-proven NASH have been evaluated [61]. After 72 weeks, treatment with semaglutide at a dose of 0.4 mg/day resulted in the histological resolution of NASH without progression of fibrosis in 59% of patients, compared with 17% in the placebo group. Furthermore, an improvement in NAS was observed in 83% of patients treated with semaglutide 0.4 mg/day, 80% of those treated with 0.2 mg/day, 71% of those treated with 0.1 mg/day and only 44% of those treated with placebo.

Although not fully clarified, the mechanism of action of GLP-1 RAs on fatty liver disease may be multifactorial. If the reduction in body weight and the improvement of glycemic control are crucial factors in determining the effects of GLP-1 on NAFLD, direct pharmacological mechanisms could also be involved. Indeed, in liraglutide-treated mice, the reduction in hepatic steatosis and endoplasmic reticulum oxidative stress occurred regardless of body weight loss (see below) [91,92].

### 3.3. Effects of GLP-1 RAs on Liver Fibrosis

Liver fibrosis is the main determinant of clinical outcomes in patients with NASH [93,94]. In particular, clinically relevant fibrosis is detectable in about 15% of diabetic patients [95]. A treatment for NAFLD must be able to impact liver fibrosis, aiming to reduce the rates of mortality, liver transplantation and liver-related events.

To date, few studies that evaluated the efficacy of GLP-1 RAs on NAFLD-related liver fibrosis are available [56,61,81,96,97,98] (Table 4). Moreover, most of the data are obtained from indirect markers of fibrosis, such as the APRI (AST-to-platelet count ratio index), the NFS (NAFLD fibrosis score), the FIB-4 (fibrosis index based on 4 factors) or from the assessment of liver stiffness (LSM). In this regard, Ohki et al. [97] observed a significant reduction in APRI in patients treated with liraglutide, similar to those treated with pioglitazone. Recently, Colosimo et al. [81] confirmed a significant reduction in FIB-4 in diabetic patients treated with GLP-1 RAs, similar to those treated with SGLT2i. Finally, Tan et al. [98] found that diabetic patients treated with liraglutide showed a significant reduction in NFS, FIB-4 and LSM after a 12-month follow-up compared to patients treated with other hypoglycemic agents. 

Few studies evaluated the impact of GLP-1 RAs on histologically detected liver fibrosis [56,61,96]. Significant data emerged in this regard from the previously discussed trial by Armstrong et al. [56]. In addition to the reduction in body weight, improvement in liver injury rates and resolution of steatohepatitis in 39% of cases, treatment with liraglutide was associated with lower progression of liver fibrosis compared to placebo. Indeed, despite the short duration of the trial (48 weeks), progression of fibrosis was detected in 9% of patients treated with liraglutide and 36% of patients treated with placebo (*p* = 0.04). However, patients with more severe fibrosis at baseline appear to have a lower response to liraglutide. Thus, early treatment of NASH with GLP-1 RAs may be more effective. Recently, Newsome et al. [61] carried out a 72-week trial on the effects of semaglutide on NASH assessed by liver biopsy before and after treatment. If semaglutide at a dosage of 0.4 mg/day leads to resolution of steatohepatitis in 59% of cases, no significant difference emerged in the improvement of fibrosis compared to placebo. However, the treatment was shown to be effective in reducing the progression of fibrosis (5% vs. 19% of the placebo group), in accordance with what previously demonstrated [56].

The protective role on fibrosis progression demonstrated by GLP-1 RAs could be explained by the effect in reducing the expression of three out of five collagen genes [99]. In fact, this effect demonstrated on mouse models exposed to semaglutide exclusively impacts the de novo production of collagen and does not affect the pre-treatment liver fibrosis. In this regard, in our opinion, it can be assumed that longer-lasting treatments could also show a significant effect on the regression of pre-existing NASH-related fibrosis. In fact, if the drug-induced reduction in hepatic fat content and the consequent lipotoxicity can determine a resolution of the steatohepatitis in the medium-term, avoiding the progression of liver damage, longer times are probably needed for the clearance of the pre-existing fibrosis.

Finally, preclinical data suggest that treatment with GLP-1 RAs could result in lower incidence of NASH-related hepatocellular carcinoma (HCC) [100]. In mouse models, liraglutide has been shown to completely suppress hepatocarcinogenesis. The protective effect on the development of HCC could be mediated both by the improvement of steatohepatitis (steatosis, inflammation, and hepatocyte ballooning) and the better glycemic compensation and weight loss, with reduced systemic inflammatory state and oxidative stress [101,102]. Moreover, in conditions of insulin resistance (such as during NAFLD), a significant imbalance in the levels of adipokines is determined, with lower levels of adiponectin and an increase in those of leptin [103]. It has been demonstrated that this imbalance favors hepatocarcinogenesis [104]. In particular, leptin promotes the proliferation and migration of neoplastic cells. GLP-1 RAs are able to increase adiponectin levels and restore the balance among adipokines [82,103,105]. This pathway could represent one of the protective mechanisms against HCC development exerted by GLP-1 RAs.

## 4. Potential Mechanisms of Action of GLP-1 RAs in NAFLD

As discussed, GLP-1 RAs have been shown to improve liver enzymes and hepatic fat accumulation as well as promote resolution of steatohepatitis in patients with NAFLD [106]. The reduction in body weight and the improved glycemic control resulting from these drugs are certainly crucial factors in the improvement of NAFLD parameters. Regardless of the mechanism, in fact, a weight loss of ≥7–10% seems to significantly improve the hepatic necro-inflammation and fibrosis [66,67]. In this regard, the previously cited meta-analysis by Armstrong et al. [73] showed that the effect of GLP-1 RAs (in particular of liraglutide) in reducing the levels of liver damage biomarkers is closely related to the body weight loss and better glycemic control. Other data also support this correlation [80]. However, it is not known whether these effects can also be attributed to a direct action of GLP-1 RAs. In fact, their use seems to exert a hepato-protective and anti-steatogenic action by inhibiting cell death and stimulating lipolysis [107]. In this regard, the GLP-1 receptor has also been identified in hepatocytes and could play a role in lowering the intrahepatocyte accumulation of fatty acids obtained through treatment with GLP-1 RAs [83,108,109,110]. Its expression appears lower in patients with NASH [110]. However, data on hepatic localization of GLP-1R are not univocal [87].

Several mechanisms have been hypothesized to explain how GLP-1 RAs could directly reduce hepatocyte storage of triglycerides, regardless of body weight loss (Figure 1). In fact, GLP-1 RAs have been shown to improve hepatic glucose metabolism and promote a reduction in lipogenesis and an increase in fatty acid oxidation [111].

Firstly, endogenous GLP-1 and GLP-1 receptor agonists could improve hepatic insulin resistance [92]. In the case of hepatic steatosis, insulin signaling pathways are largely deficient. In particular, phosphorylation of AKT, a key effector for insulin signaling, appears to be significantly decreased, hindering the post-receptor signaling cascade downstream of insulin receptor substrate-2 (IRS-2) [112]. GLP-1 ligands are able to promote the phosphorylation of AKT and other crucial molecules (PDK-1 and PKC-ζ) downstream of IRS-2 [83]. The activation of the post-receptor signaling cascade induced by GLP-1 and GLP-1 receptor agonists and the consequent modulation of insulin signaling pathway would reduce storage of hepatocyte triglyceride and consequently improve hepatic steatosis [83]. With regard to the mechanisms of insulin resistance, adipokines play a significant role. Among these, adiponectin is a hormone able to exert an insulin-sensitizing effect by binding to its receptors and to activate post-receptor signaling pathways mainly mediated by adenosine monophosphate dependent kinase (AMPK) and by peroxisome proliferator-activated receptor (PPAR)-α [113]. Adiponectin levels are markedly reduced in cases of obesity and fatty liver disease [103]. It has been hypothesized that GLP-1 RAs could increase the levels of adiponectin, improving the degree of insulin resistance, with consequent reduction in liver fat accumulation [82,103,105] and a protective role against the development of fibrosis [114]. In fact, GLP-1 increases the cyclic adenosine monophosphate (cAMP) in hepatocyte, resulting in the phosphorylation of AMPK and inhibition of lipogenesis [110,115]. However, these hypotheses need further investigation. In fact, the rise in adiponectin levels detected after treatment with GLP-1 RAs could be associated with the reduced body weight and inflammation, as with the improved glycemic control, regardless of the mechanisms involved [103].

A non-marginal role in the anti-steatotic mechanisms of GLP-1 RAs could be played by the farnesoid X receptor (FXR). This is a multifunctional receptor with a crucial role in the feeding state response, able to regulate glucose and lipid metabolism, bile acid homeostasis, and liver regeneration [116]. Recently, Errafii et al. [117] showed on an in vitro steatosis model (HepG2 cells) that the improvement of steatosis induced by GLP-1 RAs could be explained in part by the activation of the farnesoid X receptor/retinoid X receptor (FXR/RXR) pathway via direct stimulation of the GLP-1R, resulting in the inhibition of de novo lipogenesis. Furthermore, FXR activation seems to reduce the degree of fatty liver disease [118] by increasing fatty acid oxidation [119] and inhibiting PPAR-γ expression [120]. The latter appears overexpressed in NAFLD conditions and is directly associated with the development and progression of hepatic steatosis [121]. In this regard, liraglutide has been shown to reduce the hepatic expression of PPAR-γ and, consequently, the intrahepatocyte accumulation of fatty acids, restoring the physiological lipid homeostasis [122,123]. In addition, GLP-1 RAs would also promote lipolysis and fatty acid oxidation through the overexpression of family with sequence similarity 3 member A (FAM3A) [124]. Together with the reduction in steatosis via FXR, the activation of the liver X receptor/retinoid X receptor (LXR/RXR) pathway using GLP-1 RAs would account for the lower degree of liver inflammation [117]. In fact, LXR is a cholesterol sensor, with a central role in the inflammatory control and in the regulation of lipid and glucose metabolism, overexpressed in case of hepatic steatosis [125]. LXR activation has been shown to improve hepatic inflammation and reduce liposaccharides-induced liver injury in NASH mouse models [126,127,128].

A further protective mechanism that could explain the effects of GLP-1 RAs in NAFLD is represented by the potential role in preventing hepatocyte apoptosis [91]. In fact, the hepatocyte fat accumulation is able to induce endoplasmic reticulum (ER) stress and, consequently, hepatocytes apoptosis and progression of liver damage. The mechanisms of lysosomal degradation and autophagy are crucial to remove potentially toxic and protective fatty acids against ER stress-induced cell death [129,130,131]. Sharma et al. [91] demonstrated that GLP-1R stimulation is able to activate autophagy mechanisms and mitigate ER-stress mediated apoptosis, reducing the intrahepatocyte accumulation of fatty acids. This could potentially prevent the progression of liver damage in NAFLD. According to Fang et al. [108], the stimulation of GLP-1R in the liver causes a reduction in fatty acids accumulation through autophagy-dependent lipid degradation and an improvement of lysosomal function.

In addition to the potential direct mechanisms of action of GLP-1 RAs in hepatocyte, several evidences confirmed the role of these drugs in reducing the degree of low-grade chronic and systemic inflammation, a feature of the metabolic syndrome. Markers of inflammation, such as serum C-reactive protein (CRP), interleukin (IL)-6 and tumor necrosis factor (TNF)-α, are, in fact, significantly elevated in patients with insulin resistance. In addition to T2DM and increased cardiovascular risk [132], CRP is predictive of NASH in patients with NAFLD and is related to the presence and severity of liver fibrosis [133]. Beyond improving the metabolic profile, GLP-1 RAs have been shown to significantly reduce serum levels of CRP [80,134,135,136], IL-6 [137] and TNF-α [136]. In particular, treatment with semaglutide achieved a dose-dependent reduction of 25–43% in CRP values, significantly greater than with placebo [80]. Treatment with GLP-1 RAs is also able to reduce macrophage infiltration of adipose tissue and inhibit inflammatory pathways in adipocytes, concurring to ameliorate insulin sensitivity [138]. Improved systemic inflammatory status could represent a significant cofactor in reducing the risk of NAFLD progression and HCC development, as well as a crucial determinant of the cardiovascular protection exerted by GLP-1 RAs [139].

Furthermore, GLP-1 RAs have recently been shown to improve sarcopenic obesity in obese mice by influencing the skeletal muscle metabolism [140,141,142]. In addition to weight loss, semaglutide has been shown to lower the intramuscular accumulation of fat and promote protidosynthesis in these mouse models, increasing both mass and muscle function [140]. Furthermore, GLP-1 RAs have been shown to improve muscle wasting both by suppressing myostatin expression and promoting the expression of myogenic factors, as well as through anti-inflammatory and anti-apoptotic effects [141,142]. In light of the bidirectional relationship between NAFLD and sarcopenia and the significantly higher cardiovascular risk in obese sarcopenic patients compared to non-sarcopenic ones [143], GLP-1 RAs could guarantee additional direct benefits independent from weight loss and glycaemic control.

## 5. Future Perspectives

GLP-1 RAs have been shown to promote significant weight loss, reduce liver injury indices and steatosis, induce resolution of steatohepatitis, and slow the progression of fibrosis in patients with NASH. However, the potential of GLP-1 receptor agonists in NASH has not been fully explored. In fact, long-term trials that aim to evaluate the efficacy of GLP-1 RAs in reducing liver fibrosis are not currently available. If the improvement of lipotoxicity and the inhibition of de novo collagen production occur soon after starting treatment with GLP-1 RAs, the regression of pre-existing fibrosis could, in fact, require longer observation times and longer-lasting treatment. These indications can be borrowed from the experience acquired in post-viral hepatitis or cirrhosis. After eradication of HCV infection or viremic suppression in HBV infection, only a long follow-up was able to demonstrate significant regression of liver fibrosis [144,145,146]. Moreover, the long-term suppression of NASH and the favorable anti-oncogenic pathways exhibited by the GLP-1 RAs treatment could reduce the risk of de novo HCC and recurrence [100,139]. Specific trials about the impact of GLP-1 RAs treatment on the risk of HCC incidence or recurrence are currently lacking. However, at present. The available evidence on the benefits of GLP-1 Ras allows us to hypothesize a potential efficacy of this class of drugs in achieving long-term regression of fibrosis and a reduced incidence/recurrence of HCC in patients with NASH. These hypotheses need to be verified in dedicated trials.

Due to the complex and heterogeneous pathophysiology of NASH, it has been hypothesized that the simultaneous action on different pathways could exert a synergistic effect, optimizing therapeutic results. In this perspective, the possibilities of double (GLP-1 and GIP or GLP1 and glucagon) or triple (GLP-1, GIP and glucagon) receptor agonism [147,148] are the issue of recent research. Glucose-dependent insulinotropic peptide (GIP) acts by increasing insulin secretion, hepatic glucose and triglyceride uptake in adipose tissue and liver as well as reducing gluconeogenesis and glucagon secretion [149]. Glucagon is also involved in glucose and lipid homeostasis, as well as in the regulation of energy expenditure and food intake [17]. In particular, it inhibits hepatic synthesis and promotes beta-oxidation of fatty acids [148,150] and, probably, lipolysis in adipose tissue [151]. It also increases energy expenditure by inducing thermogenesis in brown adipose tissue [152] and reduces calorie intake [153]. However, glucagon counteracts the insulin action, promoting gluconeogenesis and hepatic glycogenolysis [148]. Since the activation of these pathways sometimes produces opposite effects, the concurrent action on two or more of them could result in a complementary and synergistic effect [154]. Dual and triple agonists have been shown to promote greater body weight loss and reduce food intake in obesity animal models than GLP-1 RAs alone [17,155,156,157,158]. Moreover, a dual/triple therapy could theoretically allow one to reduce GLP-1 RAs dosage and, consequently, gastrointestinal adverse effects, enhancing treatment compliance [159].

Increasing evidence is available on the dual agonisms of GLP-1 and glucagon [158,160,161,162]. Both glucagon and GLP-1 reduce appetite and promote weight loss; additionally, glucagon also increases energy expenditure. Moreover, in dual agonisms, GLP-1 could have a protective role against the hyperglycemic effect of glucagon. In mice and non-human primates, dual GLP-1R/GCGR (glucagon receptor) agonists have been shown to induce excellent glycemic control and significantly greater weight loss than liraglutide monotherapy, increasing energy expenditure and reducing total energy intake by 60–70% [158]. In human volunteers, the co-administration of GLP-1 RAs and GCG RA appears to be able to reduce food intake and increase energy expenditure compared to single administration of the two drugs [160]. Furthermore, in a randomized, placebo-controlled, double-blind, phase 2a trial, Ambery et al. [161] showed that GLP-1R/GCGR agonist MEDI0382 significantly reduces blood glucose levels and body weight in obese/overweight T2DM patients.

Similar to GLP-1/GCG dual agonism, the co-activation of GLP-1 receptor and GIP represents a fascinating perspective in the treatment of T2DM and obesity [17]. In fact, if GIP receptor agonists (GIP RA) are effective in stimulating insulin secretion and favoring peripheral glucose uptake, alone they are unable to influence body weight [17,163]. However, it has been shown in animal models that GIP RAs potentiates GLP-1-induced weight loss and reduces food intake [164,165]. In obese and diabetic rodents, co-administration of GLP-1/GIP receptor agonists (GLP-1 RA/GIP RA) seems to enhance the antihyperglycemic and insulinotropic efficacy of selective GLP-1 RAs and promote a reduction in fat mass [163]. In humans, once-weekly subcutaneous dual GLP-1R/GIPR agonist LY3298176 administration obtained greater weight loss and glucose control than dulaglutide, with an acceptable tolerability profile [166]. After 26 weeks of treatment, LY3298176 reduced glycosylated hemoglobin levels to <6.5% in up to 82% of cases (compared to 39% with dulaglutide and 2% with placebo), favored a reduction of at least 5% in body weight up to in 71% of cases (compared to 22% with dulaglutide and 0% with placebo) and at least 10% up to 39% of cases (compared to 9% with dulaglutide and 0% with placebo), and resulted in a reduction in circumference waist up to −10.2 cm (compared to −2.5 cm with dulaglutide and −1.3 cm with placebo). Moreover, dual GLP-1 RA/GIP RA tirzepatide was recently confirmed to be superior to both insulin degludec and semaglutide in reducing glycosylated hemoglobin levels and body weight at all doses tested [167,168]. In particular, the differences between 15-mg tirzepatide and the 1-mg semaglutide were −0.45 percentage points in glycosylated hemoglobin levels reduction and −5.5 kg in body weight reduction (*p* < 0.001) [167]. The dual GLP-1/GIP receptor agonism could, therefore, have synergistic effects. However, the mechanism through which GIP RAs are able to potentiate the effects of GLP-1 RAs is not known. In this regard, tirzepatide has been shown to induce weight-independent insulin sensitization in obese mice [169]. In GlP-1 R-null mice, tirzepatide improved insulin sensitivity by enhancing glucose disposal in white adipose tissue in the absence of GLP-1 RA-induced weight loss. The dual receptor agonism could, therefore, counteract the two pathogenetic mechanisms of T2DM (adiposity-induced insulin resistance and pancreatic insulin deficiency) more effectively than selective mono-agonists [163].

If double receptor agonism GLP-1/GCG and GLP-1/GIP have shown a synergistic action on glycemic control and weight loss, triple GLP-1/GCG/GIP agonism could lead to further maximization of results. In fact, the concomitant action on three key metabolic pathways could be more effective in reducing body weight and in the treatment of obesity, promoting the reduction in food intake (GLP-1, glucagon), the increase in energy expenditure (glucagon) and in peripheral glucose uptake with potentiation of GLP-1-induced weight loss (GIP), mitigating the hyperglycemic effect of glucagon. To date, however, no studies performed in humans are available. In mouse models, triple agonists were found to be superior to both monoagonists and double agonists in reducing body weight and glycemic control [170,171,172].

While dual and triple agonists could have a potential efficacy in the treatment of T2DM and obesity, few data are currently available on their role in NAFLD. In this regard, Finan et al. [171] preliminarily showed the excellent efficacy of triple agonists in reversing fatty liver disease in mouse models. This action appears significantly superior to both monoagonists and double agonists. The effectiveness of unimolecular peptide triagonists in reversed diet-induced steatohepatitis in mice was then confirmed by Jall et al. [170]. More recently, the combination of GLP-1 Ras with GCG RA or with GIP RA in NASH murine models allowed one to achieve additional weight loss and, more importantly, to determine a greater improvement in histological NAFLD activity score compared to treatment with mono-agonists [172]. Furthermore, dual receptor agonists GLP-1/GCG and the triple combination of mono-agonists led to a significantly greater reduction in the histological non-alcoholic fatty liver disease activity score compared to high-dose liraglutide, despite similar weight loss. To our knowledge, only one trial on the efficacy of the combination of multiple agonists in the treatment of NAFLD in humans is available to date. Nara et al. [173] evaluated the effects of cotadutide, a dual GLP-1 and glucagon receptor agonist, in 834 overweight/obesity and uncontrolled T2DM patients. Through a phase 2b trial, cotadutide 300 μg was shown to significantly improve liver injury enzymes, propeptide of type III collagen level, FIB-4, and NAFLD fibrosis score compared to placebo (but not compared to liraglutide) and reduce body weight more than treatment with liraglutide 1.8 mg. These data support further evaluation of cotadutide in NASH treatment.

In order to maximize the results obtained using GLP-1 RAs in NASH, other combination therapies have also been hypothesized and tested. Specifically, the efficacy of coadministration of GLP-1 RAs and FXR agonist obeticholic acid in mice was evaluated [174]. Combination therapy resulted in a significantly greater reduction in body weight and, above all, in liver steatosis, inflammation, and fibrosis compared to monotherapies, confirming the complementary or synergistic effect of the association between the agonism of GLP-1 and FXR in NASH treatment.

## 6. Conclusions

GLP-1 receptor agonists have been shown to be effective in reducing body weight, liver injury indices, and liver fat content. Several evidences also suggest that these drugs are able to promote the resolution of steatohepatitis in a non-negligible proportion of patients with NASH and to reduce progression of hepatic fibrosis. No evidence is currently available on the efficacy of GLP-1 RAs in improving pre-existing liver fibrosis in patients with NAFLD. However, data after long-term treatment with GLP-1 RAs are not yet available. Additional benefits are expected from double and triple agonists, but specific human clinical trials are needed.

## Figures and Tables

**Figure 1 ijms-24-01703-f001:**
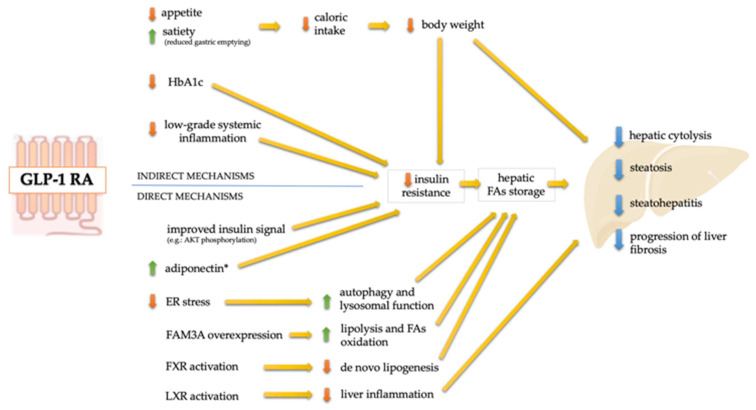
Direct and indirect mechanisms of action of GLP-1 RAs potentially involved in the NAFLD treatment. ER: endoplasmic reticulum; FAs: fatty acids; FAM3A: family with sequence similarity 3 member A; FXR: farnesoid X receptor; HbA1c: glycosylated hemoglobin; LXR: liver X receptor. * also determined by the reduction in body weight.

**Table 1 ijms-24-01703-t001:** Main evidences on the effect of liraglutide and semaglutide on weight loss.

Year	First Author	Ref.	Sample Size	Study Typology	Evaluated Drugs	Benefit	Results
2009	Astrup	[38]	564 non-diabetic obese	Prospective double-blind	Liraglutide vs orlistat vs placebo	Yes	Liraglutide resulted in greater weight loss than placebo or orlistat
2013	Wadden	[43]	422 obese/overweight	Prospective randomised	Liraglutide vs placebo	Yes	Liraglutide plus diet and exercise maintained weight loss achieved by caloric restriction and induced further weight loss
2015	Pi-Sunyer	[33]	3731 non-diabetic obese	Prospective double-blind	Liraglutide vs placebo	Yes	Liraglutide induced greater weight loss (8.4 Kg) than placebo (2.8 Kg)
2015	Davies	[47]	846 obese/overweight T2DM	Prospective double-blind	Liraglutide vs placebo	Yes	Liraglutide (3.0 mg/day), compared with placebo, resulted in greater weight loss
2016	de Boer	[41]	151 obese insulin-using T2DM	Prospective	Liraglutide or Exenatide	Yes	Liraglutide or Exenatide led to sustained weight reduction and daily insulin dose
2016	Blackman	[44]	359 non-diabetic obese	Prospective double-blind	Liraglutide vs placebo	Yes	Liraglutide (3.0 mg/day) plus lifestyle therapy led to a greater weight loss than placebo plus lifestyle therapy
2016	Armstrong	[56]	52 NASH patients	Multicentre, double-blinded	Liraglutide vs placebo	Yes	Liraglutide 1.8 mg/day led to a greater weight loss than placebo
2017	Halawi	[45]	40 obese patients	Prospective double-blind	Liraglutide vs placebo	Yes	Liraglutide delays gastric emptying and reduces body weight more than placebo
2017	Petit	[57]	68 uncontrolled T2DM patients	Prospective single-center	Liraglutide	Yes	Liraglutide 1.2 mg/day significantly reduced body weight
2018	Frøssing	[58]	72 obese/overweight with PCOS	Prospective double-blind	Liraglutide vs placebo	Yes	Liraglutide results in greater weight loss than placebo
2019	Feng	[59]	85 T2DM and NAFLD patients	Prospective, randomized	Liraglutide vs gliclazide	Yes	Liraglutide results in greater weight loss than gliclazide
2019	Yan	[50]	75 NAFLD and metformin-uncontrolled T2DM patients	RCT	Liraglutide vs sitagliptin vs insulin glargine	Yes	Combined with metformin, both liraglutide and sitagliptin, but not insulin glargine, reduced body weight
2020	Kelly	[40]	251 obese adolescents	Prospective double-blind	Liraglutide vs placebo	Yes	Liraglutide (3.0 mg/day) plus lifestyle therapy led to a greater weight loss than placebo plus lifestyle therapy in adolescent with obesity
2020	Wadden	[42]	282 obese	Prospective double-blind	Liraglutide vs placebo	Yes	Liraglutide (3.0 mg/day) amplifies weight loss due to intensive behavioral therapy
2020	Garvey	[48]	396 obese/overweight and insulin-treated T2DM patients	Prospective double-blind	Liraglutide vs placebo	Yes	Liraglutide (3.0 mg/day) led to greater weight loss than placebo
2017	Blundell	[39]	28 obese	Prospective double-blind	Semaglutide vs placebo	Yes	Semaglutide resulted in lower ad libitum energy intake and greater weight loss than placebo
2018	O’Neil	[31]	957 non-diabetic obese	Prospective double-blind	Semaglutide vs Liraglutide	Yes	Semaglutide (≥0.2 mg/day) resulted in greater weight loss than liraglutide (any dose)
2018	Pratley	[32]	1201 T2DM patients	Prospective randomised	Semaglutide vs Dulaglutide	Yes	Semaglutide 0.5 mg/week resulted in greater weight loss than dulaglutide 0.75 mg/week. Semaglutide 1 mg/week resulted in greater weight loss than dulaglutide 1.5 mg/week
2019	Matikainen	[60]	22 controlled T2DM patients	Prospective single-blind	Liraglutide vs placebo	No	Similar weight loss between 16-week liraglutide 1.8 mg/day and placebo group
2021	Wilding	[34]	1961 non-diabetic obese	Prospective double-blind	Semaglutide vs placebo	Yes	Semaglutide (2.4 mg/week) plus lifestyle intervention resulted in a greater reduction in BMI than lifestyle intervention alone
2021	Davies	[35]	1210 insulin-untreated diabetic obese/overweight patients	Prospective double-blind	Semaglutide 2.4 vs 1 mg/week vs placebo	Yes	Semaglutide 2.4 mg/week resulted in a greater weight loss than Semaglutide 1.0 mg/week or placebo
2021	Wadden	[36]	611 non-diabetic obese/overweight patients	Prospective double-blind	Semaglutide vs placebo	Yes	Semaglutide (2.4 mg/week) plus lifestyle intervention resulted in a greater reduction in BMI than lifestyle intervention alone
2021	Rubino	[37]	902 non-diabetic obese/overweight patients	Prospective double-blind	Semaglutide vs placebo	Yes	Semaglutide (2.4 mg/week) plus lifestyle intervention resulted in a greater reduction in BMI than lifestyle intervention alone
2021	Newsome	[61]	320 obese/overweight NASH patients	Prospective double-blind	Semaglutide vs placebo	Yes	Semaglutide treatment resulted in a greater reduction in BMI than placebo
2022	Rubino	[46]	338 non-diabetic obese/overweight patients	Prospective randomised	Semaglutide vs Liraglutide	Yes	Semaglutide (2.4 mg/week) resulted in greater weight loss than liraglutide (3 mg/day)
2022	Weghuber	[49]	201 obese adolescents	Prospective double-blind	Semaglutide vs placebo	Yes	Semaglutide (2.4 mg/week) plus lifestyle intervention resulted in a greater reduction in BMI than lifestyle intervention alone

BMI: body mass index; BW: body weight; NA: not available; NAFLD: non-alcoholic fatty liver disease; NASH: non-alcoholic steatohepatitis; PCOS: polycystic ovary syndrome; RCT: randomized clinical trial; T2DM: type 2 diabetes mellitus.

**Table 2 ijms-24-01703-t002:** Main evidences on the effect of GLP-1 RA on liver injury indices.

Year	First Author	Ref.	Sample Size	Study Typology	Evaluated Drugs	Benefit	Results
2007	Buse	[69]	283 T2DM patients	Multicenter, double-blind trials	Exenatide vs placebo	Yes	Exenatide resulted in progressive reduction in weight and improvements in hepatic injury biomarkers
2008	Klonoff	[70]	217 T2DM patients	Open-label clinical trial	Exenatide vs placebo	Yes	Exenatide significantly reduced liver enzymes vs placebo
2011	Sathyanarayana	[71]	21 T2DM patients	NS	Exenatide vs pioglitazone	Yes	Hepatic injury biomarkers were significantly decreased by Exenatide and Pioglitazone. The reduction in ALT was significantly greater following combined therapy.
2013	Fan	[72]	117 T2DM and NAFLD patients	Prospective randomized trial	Exenatide vs metformin	Yes	Exenatide treatment results in significant improvement in liver enzymes compared with metformin.
2013	Armstrong	[73]	4442 T2DM patients	Meta-analysis	Liraglutide vs placebo	Yes	Liraglutide 1.8 mg/day reduced ALT vs placebo with a dose-dependent effect: no significant differences vs placebo with liraglutide 0.6 or 1.2 mg/day
2014	Blaslov	[74]	125 T2DM patients	Open label parallel-group uncontrolled	Exenatide vs other oral hypoglycemic agents	Yes	Exenatide results in reduction of NAFLD marker levels and intrahepatic fat quantity calculated by FLI
2014	Gluud	[75]	1070 T2DM patients with increased ALT	Systematic review	Lixisenatide	Yes	Lixisenatide increases the proportion of obese or overweight T2DM patients who achieve normalisation of ALT
2015	Eguchi	[76]	19 NASH patients	Prospective uncontrolled	Liraglutide	Yes	Liraglutide significantly improved liver enzymes in NASH patients
2016	Armstrong	[56]	52 NASH patients	Multicentre, double-blinded, phase 2 trial	Liraglutide vs placebo	Yes	Liraglutide 1.8 mg/day led to significative ALT reduction vs placebo
2017	Feng	[77]	87 T2DM and NAFLD patients	RCT	Liraglutide vs gliclazide vs metformin	Yes	Liraglutide and metformin reduce ALT and liver fat content
2017	Seko	[78]	15 biopsy-proven NAFLD and T2DM patients	Retrospective	Dulaglutide	Yes	Dulaglutide significantly reduced liver enzymes
2018	Cusi	[79]	1499 T2DM and NAFLD patients	Placebo-controlled clinical trial: post hoc analysis	Dulaglutide vs placebo	Yes	Dulaglutide significantly reduced liver enzymes vs placebo
2019	Newsome	[80]	4254 T2DM and/or obesity patients	Post hoc analysis	Semaglutide	Yes	Semaglutide significantly reduced ALT
2021	Colosimo	[81]	637 T2DM patients	Retrospective	DPP-4i vs GLP-1 RA vs SGLT-2i vs others	Yes	GLP-1 RA and SGLT-2i improve biomarkers of liver injury
2021	Newsome	[61]	320 obese/overweight NASH patients	Prospective double-blind	Semaglutide vs placebo	Yes	Semaglutide treatment resulted in a greater dose-dipendent reduction in liver enzymes than placebo

ALT: alanine aminotransferase; DPP-4i: dipeptidyl peptidase-4 inhibitors; FLI: fatty liver index; GLP-1 RA: Glucagon-like peptide-1 receptor agonists; NAFLD: non-alcoholic fatty liver disease; NASH: non-alcoholic steatohepatitis; NS: not specified; RCT: randomized clinical trial; SGLT2i: Sodium-glucose Cotransporter-2 inhibitors; T2DM: type 2 diabetes mellitus.3.3. Effects of GLP-1 RAs on Liver Steatosis

**Table 3 ijms-24-01703-t003:** Main evidences on the effect of GLP-1 RA on liver steatosis.

Year	First Author	Ref.	Sample Size	Study Typology	Evaluated Drugs	Benefit	Results
2009	Jendle	[86]	314 uncontrolled T2DM patients	Randomized, double-blind, parallel-group	Metformin + Liraglutide (0.6, 1.2 or 1.8 mg/day) or glimepiride or placebo	Yes	Liraglutide (with or without metformin) significantly reduced fat mass and fat percentage vs glimepiride
2011	Sathyanarayana	[71]	21 T2DM patients	NS	Exenatide vs pioglitazone	Yes	The addition of exenatide to pioglitazone is associated with a greater reduction in hepatic fat content as compared to pioglitazone monotherapy
2012	Cuthbertson	[87]	25 obese, T2DM and NAFLD patients	Prospective single arm	Exenatide	Yes	Exenatide significantly decreased liver fat (evaluated by MR)
2014	Braslov	[74]	125 T2DM patients	Open label parallel-group uncontrolled	Exenatide vs other oral hypoglycaemic agents	Yes	Exenatide induces greater improvement in FLI than other oral hypoglycaemic agents
2015	Tang	[85]	35 uncontrolled T2DM patients	RCT	Liraglutide vs insulin glargine	No	Twelve-week liraglutide treatment liraglutide does not modify liver proton density fat fraction
2015	Eguchi	[76]	19 NASH patients	Prospective uncontrolled	Liraglutide	Yes	Liraglutide significantly improved histological features in NASH
2016	Dutour	[88]	44 obese T2DM patients	RCT	Exenatide vs other hypoglycaemics	Yes	Exenatide induced a significant reduction in hepatic triglyceride content compared with the reference treatment
2016	Smits	[84]	52 overweight T2DM patients	Randomised, placebo-controlled	Liraglutide 1.8 mg vs sitagliptin vs placebo	No	Twelve-week liraglutide treatment does not reduce hepatic steatosis
2016	Vanderheiden	[89]	71 T2DM patients requiring high-dose insulin treatment	Single-center, randomized, double-blind	Liraglutide vs placebo	Yes	Liraglutide significantly decreased liver fat (evaluated by MR) vs placebo
2016	Armstrong	[56]	52 NASH patients	Multicentre, double-blinded, phase 2 trial	Liraglutide vs placebo	Yes	Liraglutide 1.8 mg/day led to histological resolution of NASH in 39% of patients
2017	Bouchi	[90]	17 insulin-treated T2DM patients	Randomized, open-label	Liraglutide + insulin vs insulin	Yes	Liraglutide reduces visceral adiposity and hepatic fat accumulation
2017	Feng	[77]	87 T2DM and NAFLD patients	RCT	Liraglutide vs gliclazide vs metformin	Yes	Liraglutide causes a greater reduction of intrahepatic fat than gliclazide
2017	Petit	[57]	68 uncontrolled T2DM patients	Prospective single-center	Liraglutide	Yes	Liraglutide 1.2 mg/day significantly reduced liver fat content by body weight reduction
2018	Frøssing	[58]	72 obese/overweight patients with PCOS	Prospective double-blind	Liraglutide vs placebo	Yes	Liraglutide treatment reduced liver fat content by 44% (evaluated by MR spectroscopy) compared with placebo
2019	Yan	[50]	75 NAFLD and metformin-uncontrolled T2DM patients	RCT	Liraglutide vs sitagliptin vs insulin glargine	Yes	Combined with metformin, both liraglutide and sitagliptin, but not insulin glargine, reduced intrahepatic lipid and visceral adipose tissue
2021	Colosimo	[81]	637 T2DM patients	Retrospective	DPP-4i vs GLP-1 RA vs SGLT-2i vs others	Yes	GLP-1 RA and SGLT-2Is improve biomarkers of steatosis (FLI)
2021	Newsome	[61]	320 obese/overweight NASH patients	Prospective double-blind	Semaglutide vs placebo	Yes	Semaglutide treatment resulted in a significantly higher percentage of patients with NASH resolution than placebo

DPP-4i: dipeptidyl peptidase-4 inhibitors; FLI: fatty liver index; GLP-1 RA: Glucagon-like peptide-1 receptor agonists; MR: magnetic resonance; NAFLD: non-alcoholic fatty liver disease; NASH: non-alcoholic steatohepatitis; NS: not specified; PCOS: polycystic ovary syndrome; RCT: Randomized Controlled Trial; SGLT2i: Sodium-glucose Cotransporter-2 inhibitors; T2DM: type 2 diabetes mellitus.

**Table 4 ijms-24-01703-t004:** Main evidences on the effect of GLP-1 RA on liver fibrosis.

Year	First Author	Ref.	Sample Size	Study Typology	Evaluated Drugs	Benefit	Results
2010	Kenny	[96]	8 T2DM and biopsy-proven NAFLD	Prospective, open-labeled case series	Exenatide	NA	Three of eight patients show improvement in liver histology
2012	Ohki	[97]	82 NAFLD and T2DM patients	Retrospective	Liraglutide vs pioglitazone vs sitagliptin	Yes	Liraglutide or pioglitazone significantly reduce APRI
2016	Armstrong	[56]	52 NASH patients	Multicentre, double-blinded, randomized, phase 2 trial	Liraglutide vs placebo	Yes	Liraglutide led to lower biopsy proven fibrosis progression compared with placebo
2021	Colosimo	[81]	637 T2DM patients	Retrospective	DPP-4i vs GLP-1 RA vs SGLT-2i vs others	Yes	GLP-1 RA and SGLT-2Is improve biomarkers of fibrosis (FIB-4)
2021	Newsome	[61]	320 obese/overweight NASH patients	Prospective double-blind	Semaglutide vs placebo	No	No significant improvement in fibrosis stage
						Yes	Semaglutide 2.1 mg/week led to lower biopsy proven fibrosis progression compared with placebo
2022	Tan	[98]	1765 T2DM patients	Prospective cohort	Liraglutide	Yes	Liraglutide treatment is associated with decreased liver fibrosis

DPP-4i: dipeptidyl peptidase-4 inhibitors; FIB-4: fibrosis index based on 4 factors; GLP-1 RA: Glucagon-like peptide-1 receptor agonists; NAFLD: non-alcoholic fatty liver disease; NASH: non-alcoholic steatohepatitis; SGLT2i: Sodium-glucose Cotransporter-2 inhibitors; T2DM: type 2 diabetes mellitus.

## Data Availability

No new data were created.

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
