# Peer review of "GLP-1 Receptor Agonists in Non-Alcoholic Fatty Liver Disease: Current Evidence and Future Perspectives"

_ijms, 2023, doi:10.3390/ijms24021703_

Round 1

Reviewer 1 Report

Agree to accept.

1. This review mainly addressed the role of GLP-1 RA in NAFLD and described their potential impact on the course of liver disease and its complications. The authors collect more than 50 pieces of literature to show the evidence. It’s closely relevant to the topic and should be interesting to the public reading.  

2. The topic is not a new field, but the authors choose a narrow angle to describe the role of GLP-1 RA, as a sample, there is a review paper published in 2012, by Lee, Jinmi, et al. "GLP-1 receptor agonist and non-alcoholic fatty liver disease." Diabetes & metabolism journal 36.4 (2012): 262-267. Mainly discuss the mechanism of GLP-1 RA.

3. This paper is well organized and well written, and also easy to read.  

4. The conclusion emphasizes again that GLP-1 receptor agonists have been shown to be effective in reducing body weight, liver injury indices, and liver fat content. Definitely consistent with the evidence presented, and well addressed the main question posted.

Author Response

Reviewer 1

  1. This review mainly addressed the role of GLP-1 RA in NAFLD and described their potential impact on the course of liver disease and its complications. The authors collect more than 50 pieces of literature to show the evidence. It’s closely relevant to the topic and should be interesting to the public reading. 2. The topic is not a new field, but the authors choose a narrow angle to describe the role of GLP-1 RA, as a sample, there is a review paper published in 2012, by Lee, Jinmi, et al. "GLP-1 receptor agonist and non-alcoholic fatty liver disease." Diabetes & metabolism journal 36.4 (2012): 262-267. Mainly discuss the mechanism of GLP-1 RA.

We wish to thank the reviewer for the comment. Section 4 "Potential mechanisms of action of GLP-1 RA in NAFLD" is entirely devoted to the mechanisms through which GLP-1 RAs would be able to induce significant improvements in patients with NAFLD. According with the reviewer's suggestion, the review "GLP-1 receptor agonist and non-alcoholic fatty liver disease" by Lee et al. has been added to the text and among the references.

  1. This paper is well organized and well written, and also easy to read. 4. The conclusion emphasizes again that GLP-1 receptor agonists have been shown to be effective in reducing body weight, liver injury indices, and liver fat content. Definitely consistent with the evidence presented, and well addressed the main question posted.

We thank the reviewer for the comment and for appreciating our work.

Reviewer 2 Report

With a prevalence of up to 35%, NAFLD is the most prevalent liver disease in developed nations and is gradually growing in importance as a clinical issue. A weight loss of between 7 and 10 percent has been shown to be the only therapy for NAFLD so far that can significantly improve the histological grade of fibrosis, steatosis, and hepatic cytolysis. In patients with NAFLD, GLP-1 Receptor Agonists (GLP-1 RA) have a significant effect on body weight as well as clinical, biochemical, and histological markers of fatty liver and fibrosis. GLP-1 RAs may therefore be a useful tool to combat against both diabetes mellitus and NAFLD. This review's objectives are to provide an overview of the current knowledge base regarding the use of GLP-1 RAs in the management of NAFLD and to make predictions about possible future developments. This review has been meticulously crafted and comprehensively summarizes the main published studies on this subject. As a result, I would like to congratulate the authors on their efforts and recommend that this review be published.

Author Response

Reviewer 2

With a prevalence of up to 35%, NAFLD is the most prevalent liver disease in developed nations and is gradually growing in importance as a clinical issue. A weight loss of between 7 and 10 percent has been shown to be the only therapy for NAFLD so far that can significantly improve the histological grade of fibrosis, steatosis, and hepatic cytolysis. In patients with NAFLD, GLP-1 Receptor Agonists (GLP-1 RA) have a significant effect on body weight as well as clinical, biochemical, and histological markers of fatty liver and fibrosis. GLP-1 RAs may therefore be a useful tool to combat against both diabetes mellitus and NAFLD. This review's objectives are to provide an overview of the current knowledge base regarding the use of GLP-1 RAs in the management of NAFLD and to make predictions about possible future developments. This review has been meticulously crafted and comprehensively summarizes the main published studies on this subject. As a result, I would like to congratulate the authors on their efforts and recommend that this review be published.

We wish to thank the reviewer for the comments.

Reviewer 3 Report

This manuscript is a well-structured review describing the role of GLP-1 agonists in patients with NAFLD and future perspectives. I would like to make two minor comments.

1.     It is considered that the dose of Semaglutide should be revised to 2.4mg/wk. (citation 61)

2.     In the putative mechanism of NAFLD, research results in which sarcopenia and abundant visceral adiposity are also responsible for one axis have been recently reported. Please consider adding reports on the positive effects of GLP-1 agonists on muscle and adipose tissue. (FIGURE 1.)

Thank you.

Author Response

Reviewer 3

This manuscript is a well-structured review describing the role of GLP-1 agonists in patients with NAFLD and future perspectives. I would like to make two minor comments.

  1. It is considered that the dose of Semaglutide should be revised to 2.4mg/wk. (citation 61)

We wish to thank the reviewer for the comments. The dosage of semaglutide was corrected according to what was reported in the manuscript by Newsome et al. (0.1, 0.2 and 0.4 mg/day).

  1. In the putative mechanism of NAFLD, research results in which sarcopenia and abundant visceral adiposity are also responsible for one axis have been recently reported. Please consider adding reports on the positive effects of GLP-1 agonists on muscle and adipose tissue.

We wish to thank the reviewer for the comment. The role of GLP-1 RAs in ameliorating sarcopenia completes the picture of the mechanisms of action of this class of drugs and has been added in the text according to the suggestions.